# Comprehensive Analysis of the Impact of the Icing of Wind Turbine Blades on Power Loss in Cold Regions

**Zhenju Chuang, Hui Yi \*, Xin Chang \*, Hongxiang Liu, Haidian Zhang and Lulin Xia**

Naval Architecture and Ocean Engineering College, Dalian Maritime University, Dalian 116026, China; zhenjuchuang@dlmu.edu.cn (Z.C.); liuhongxiang@dlmu.edu.cn (H.L.); haidianzhang@dlmu.edu.cn (H.Z.); xialulin@dlmu.edu.cn (L.X.)
\* Correspondence: yih@dlmu.edu.cn (H.Y.); xin.chang@dlmu.edu.cn (X.C.)

**Abstract:** Blade icing often occurs on wind turbines in cold climates. Blade icing has many adverse effects on wind turbines, and the loss of output power is one of the most important effects. With the increasing emphasis on clean energy around the world, the design and production of wind turbines tend to be large-scale. So this paper selected the 15 MW wind turbine provided by NREL (American Renewable Energy Laboratory) to study the influence of blade icing on output power. In this paper, a multi-program coupled analysis method named CFD-WTIC-ILM (CFD: Computational fluid dynamics; WTIC: Wind Turbine Integrated Calculation; ILM: Ice loss method) was proposed to analyze the whole machine wind turbine. Firstly, Fensap-ice was used to simulate the icing of the wind turbine blades, and then the icing results were input into WTIC for the integrated calculation and analysis of the wind turbine. Then, the WTIC calculation results were used to simulate SCADA (supervisory control and data acquisition) data and input into ILM to calculate the power loss. Finally, this paper analyzed the comprehensive influence of icing on output power. The calculation results show that the ice mainly accumulates on the windward side of the blade. Icing has a great influence on the aerodynamic characteristics of the airfoil, leading to a significant decrease in the power curve. The rated wind speed is pushed from 10.59 m/s to 13 m/s. The power loss of the wind turbine in the wind speed optimization stage is as high as 37.48%, and the annual power loss rate caused by icing can reach at least 22%.

**Keywords:** 15 MW wind turbine; cold climate; blade icing; CFD-WTIC-ILM method; power loss

## 1. Introduction

The number of wind turbines that have been installed around the world has increased significantly in recent years. There have been enormous developments in offshore wind turbines because of their unique resource advantages over land-based turbines. Due to the low temperatures, in areas with cold climates, the phenomenon of wind turbine blade icing is common, which not only affects the service life of the wind turbines, but also seriously affects the power generation of the wind turbine. This paper focused on the effect of blade icing on the power produced by a 15 MW wind turbine provided by NREL.

Previous studies have utilized many methods to simulate the phenomenon of wind turbine icing, and the most real data can be obtained by field measurement. Linyue Gao et al. [1] conducted a wind turbine icing field test on a 50 MW wind farm. The icing conditions of the wind turbine blades were photographed by a UAV system. The icing characteristics of a 50 m blade were studied, and they evaluated the influence of icing on the working state and power generation of the wind turbine. Due to the difficulty and time-consuming nature of these field measurements, however, wind tunnel tests and CFD were proposed for use in follow-up research.

Wind tunnel tests are the most commonly used experimental method to simulate blade icing in the early stage, as these tests can simulate the icing phenomenon under

specific conditions. Peter Struk et al. [2] carried out surface ice tests on an NACA0012 airfoil and measured the drag coefficient of the airfoil after icing. Kraj et al. [3] used an ice wind tunnel test to simulate the ice-covered growth process of small wind turbine blades. Yiqiang Han et al. [4] carried out wind tunnel tests on running wind turbine blades and simulated the real icing process of these wind turbine blades. With the rapid development of science and technology, it has become common for people to use computer-based CFD methods to simulate icing. Barber et al. [5] combined a numerical simulation method with an experimental method to analyze the influence of blade icing on wind turbine power generation. These authors concluded that icing can reduce the output power of the wind turbine by 17%, and the most influential part of the blade surface icing on the power is the 5% of the blade known as the leading edge. Various studies [6–9] concluded through numerical simulations on ice-structure interaction process. The influence of glaze is greater than that of rime. Simo et al. [10] simulated the aerodynamic and mass unbalanced icing of wind turbine rotors and blades and concluded that the icing of wind turbine blades not only reduces the power generated by the turbine but also leads to increased vibration, thus reducing the turbine's service life. Xiangyi Kong et al. [11] used Fensap-ice simulation software to conduct a numerical simulation of the icing of an NACA0012 airfoil and compared the icing conditions at different temperatures.

Previous studies have used many methods to evaluate the power loss due to the icing of wind turbines, such as the blade element momentum (BEM) method, the experimental evaluation method, empirical formula inference, and data prediction, etc. From 2016 to 2019, Yirtici et al. [12–14] developed a tool to predict icing based on the blade element momentum (BEM) method for the prediction of the power loss caused by icing on wind turbine blades. Homola et al. [15] analyzed the influence of icing on the output power of an NREL 5 MW wind turbine using the blade element momentum theory and concluded that icing on the blade can lead to a power loss of about 27% in the wind speed optimization stage of the power curve. Lamraoui et al. [16] studied the influence of environmental parameters on the icing characteristics and power loss of wind turbine blades by numerical simulations. The results showed that the power loss after icing was mainly located at 0.8 along the spanwise direction of the blade, and the maximum power loss can reach 40%. Etemaddar et al. [7] concluded through simulations that ice cover causes wind turbines to produce wind speed lag of rated output power, and the maximum power loss can reach 35%. In the literature, the power loss caused by wind turbine icing is usually obtained by different calculation methods, resulting in different results. Current production loss methods usually use a constant −15% or −25% clean power curve drop as an indication of icing. This method is not reliable compared to the actual icing state of wind turbine blades. The t19-ice loss method (TIM) proposed by the International Energy Agency (IEA) in 2019 is a standard method for calculating power loss based on wind turbine SCADA data.

In this paper, the CFD-WTIC-ILM multi-program coupling analysis method proposed is a combination of a CFD numerical simulation, an integrated calculation for the wind turbine, and the ILM method. Firstly, the 15 MW wind turbine blade provided by NREL is segmented reasonably. The cross-sectional coordinate data were extracted to establish a model, and the Fensap-ice was used to simulate the icing of wind turbine blades. Then, the icing results were taken as input data to carry out the numerical modeling of the wind turbine and its associated icing, and this was input into WTIC for integration calculation and analysis. The power output value under different working conditions was preliminarily analyzed. Then, the WTIC calculation results were used to simulate the SCADA data of the wind turbine and input into the ILM method for calculation. Finally, the power comparison between the clean wind turbine blade and the blade affected by icing was carried out to analyze the power loss.

Compared with previous power research methods, the CFD-WTIC-ILM method is a complete set of power loss research methods, which is more convincing and accurate. This paper aimed to verify the feasibility of this power loss evaluation method and takes the 15 MW wind turbine as the research object. The research conclusions of this paper have a

certain reference value for determining the power loss of 15 MW or higher power wind turbines that occurs due to icing and can also provide more guidance for the subsequent research on power loss.

## 2. Numerical Simulation

### 2.1. Wind Turbine Model

The research object of this paper was the 15 MW wind turbine model provided by NREL. As this paper focused on the influence of wind turbine blade icing on power. The wind load is the main factor affecting blade icing, while wave load has little influence, so we only considered the influence of wind load on blade icing. The Monopile wind turbine can be used as the rigid constraint at the bottom of the wind turbine. To a great extent, the influence of the displacement of wind turbines caused by load on blade icing and power is reduced. The Monopile foundation structure shown in Figure 1 was selected. The blades of the 15 MW wind turbine were composed of eight airfoils in Figure 2. The important parameters of wind turbines are shown in Table 1.

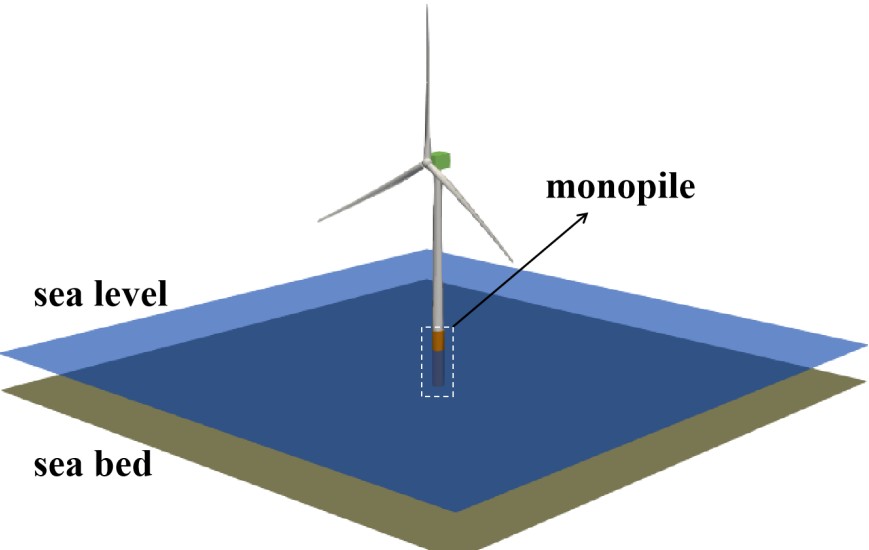

**Figure 1.** The 15 MW wind turbine model.

**Table 1.** Important parameters of the 15 MW wind turbine.

| Parameter | Value | Units |
|---|---|---|
| Rated Power | 15 | MW |
| Rotor diameter | 240 | m |
| Hub height | 150 | m |
| Blade length | 117 | m |
| Root diameter | 5.2 | m |
| Cut-in wind speed | 3 | m/s |
| Optimized wind speed | 6.98 | m/s |
| Rated wind speed | 10.59 | m/s |
| Cut-out wind speed | 25 | m/s |
| Maximum tip speed | 95 | m/s |
| Minimum rotational speed | 5 | rmp |
| Rated rotational speed | 7.5 | rmp |

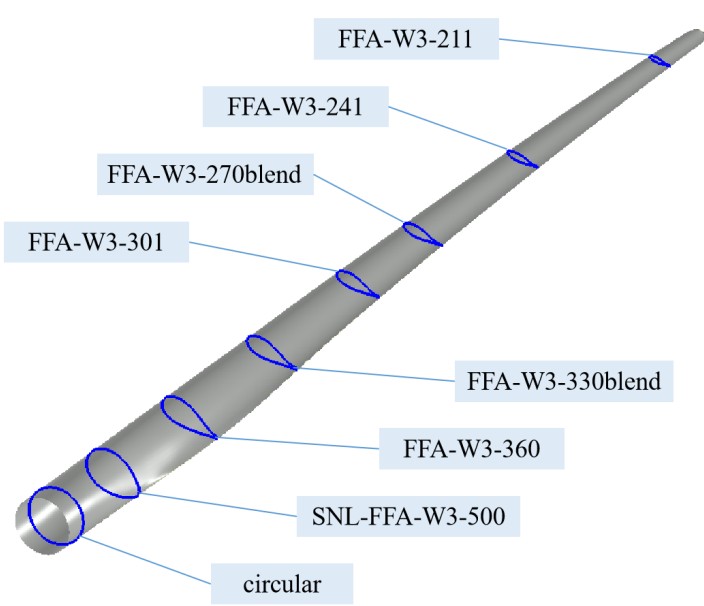

**Figure 2.** Blade section selection diagram.

*2.2. Numerical Model*

2.2.1. Icing Section Selection

The blade length of the 15 MW wind turbine is 117 m. To accurately simulate the contour shape of blade icing, we selected 13 sections for ice accumulation simulation. Lamraoui et al. [16] showed that the power loss of the blade after icing was mainly located at 0.8 along the blade span. Therefore, the choice of a cross-section from the tip to the root of the blade was arranged in a dense to sparse form, as shown in Figure 3.

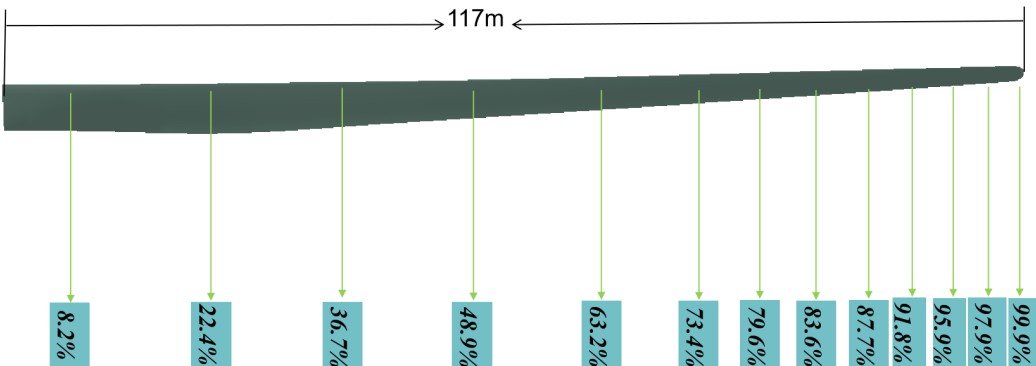

**Figure 3.** Blade section selection.

2.2.2. Icing Calculation Model

The airfoil data of the selected section were respectively imported into Catia software for the modeling of the C-type fluid domain, as shown in Figure 4. We chose the structured grid for calculation. Fensap-ice is specialized 3D ice accumulation software, so the entire C-type fluid domain needs to be stretched horizontally as shown in Figure 5 during modeling.

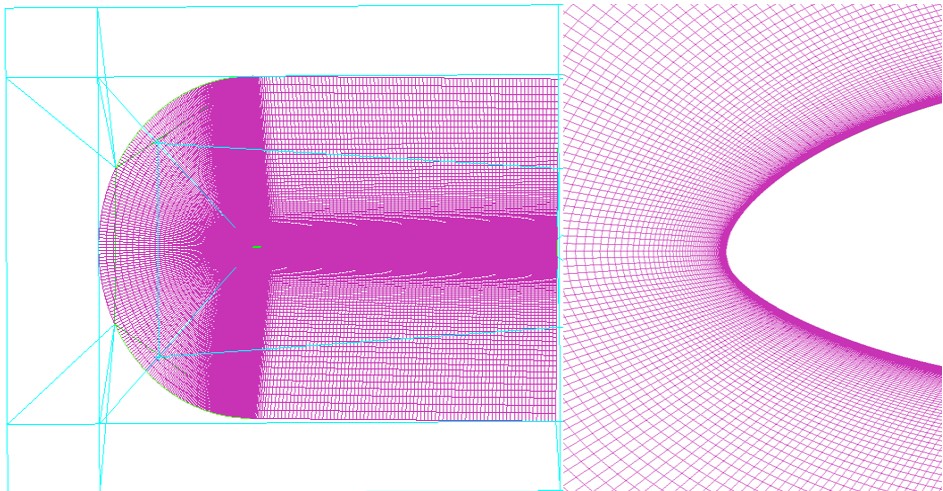

**Figure 4.** C-type fluid domain and structural grid.

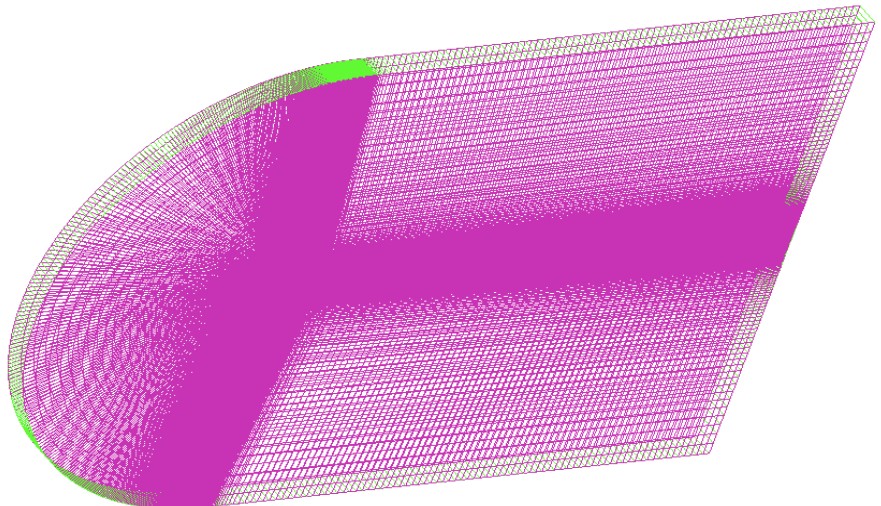

**Figure 5.** Model stretching.

### 2.2.3. Environmental Parameter

The offshore wind farm (Tahkoluoto offshore wind farm) built in Finland in 2017 is located in the northernmost Bosnian Gulf of the Baltic Sea. In this paper, the influence of blade icing on the power of the 15 MW wind turbine was analyzed based on the polar sea meteorological conditions. Table 2 shows the meteorological conditions of the Arctic ice. The characteristic length of the blade section was based on the chord length of the section at the selected position of the real blade. The radius of the C-type fluid domain was ten times the length of the chord length, the stretching length was equal to the chord length. The thickness of the first layer of the grid was calculated according to the parameters such as the combined speed of different sections. The grid division uses structured grids, and the total number of grids is much less than that of unstructured grids. The wind speed is 13 m/s, the temperature is −15 °C; the air pressure is rated atmospheric pressure, and LWC (Liquid Water Content) takes 0.2; MVD (Median Volume Diameter) takes 20 [17]; icing time 3 h.

**Table 2.** Arctic meteorological conditions.

| Parameter | Value | Units |
|---|---|---|
| Temperature | 15~0 | °C |
| Wind speed | ⩽13 | $m \cdot s^{-1}$ |
| LWC | 0.04~0.20 | $g \cdot m^{-3}$ |
| MVD | 4~20 | µm |
| Air density | 1.293 | $kg \cdot m^{-3}$ |
| Air viscosity | $1.7162 \times 10^{-5}$ | $m^2 \cdot s^{-1}$ |

## 3. Theoretical Method

### 3.1. Method and Procedure

The CFD-WTIC-ILM multi-program coupling analysis method was proposed, which requires Fensap-ice software to simulate the icing of various sections of the blade by using $k - \varepsilon$ model. Then, the aerodynamic performance of each airfoil section of the blade after icing was calculated. The WTIA module compared and calculated the blade motion and dynamic response of the wind turbine before and after icing. The ILM program used statistical methods to compare the power of the wind turbine before and after icing and to calculate the power loss. The coupling analysis of the three modules can systematically calculate the power loss of wind turbine blades due to icing. The flow chart of this method is shown in Figure 6.

The CFD-WTIC-ILM method was analyzed objectively. Its advantage lies in coupling multiple programs together to realize the cross-integration of multiple methods and theories. It also has certain limitations and shortcomings, such as: 1. The effect of two-dimensional icing on blades is different from that of three-dimensional icing; 2. The resources of SCADA data are limited, and different environmental factors have certain limitations in analyzing the power loss of wind turbines.

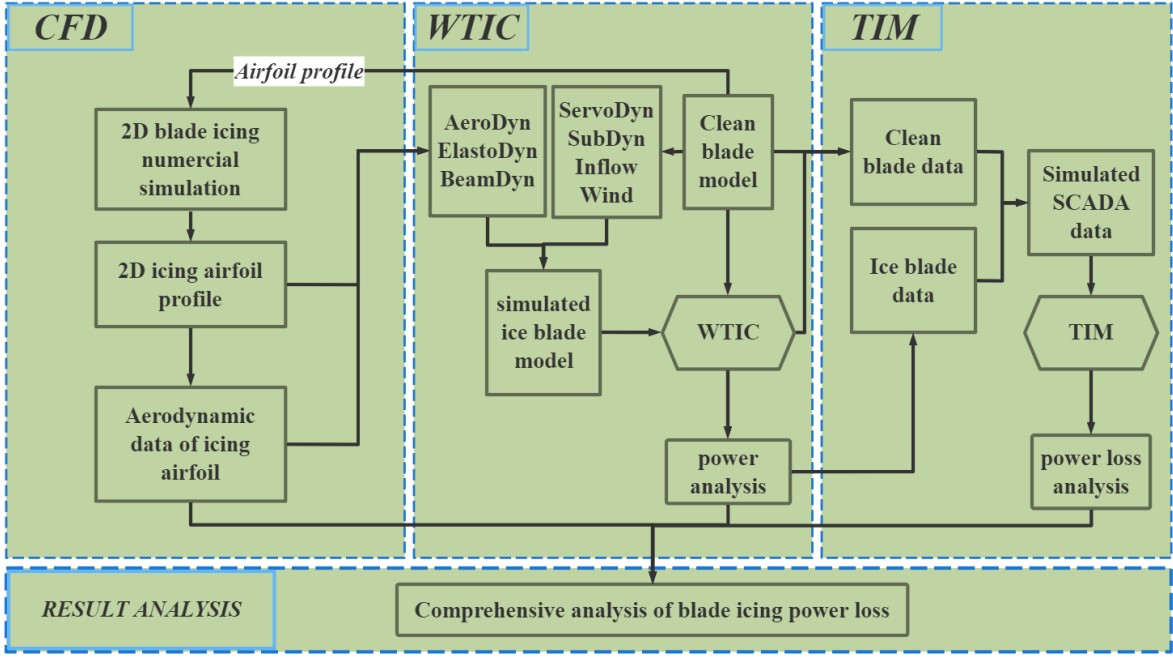

**Figure 6.** Method flow chart.

### 3.2. The Icing Growth Model

The icing of leaves is an unstable process. Ice accretion will change the geometry of the original blade airfoil, thus affecting the flow field and water droplet impact characteristics. To simulate the dynamic shape change of ice more realistically, this paper used the Fensap-ice multi-shot icing calculation method [18].

### 3.2.1. Motion Models of Air and Water Droplets

CFD is used to solve the dynamic equations of fluids and gases numerically. In this paper, Fensap-ice was used to conduct multi-phase numerical simulation, and the $N - S$ solver was used to analyze the steady-state flow field around turbine blades. Using the integrated thermo-fluid dynamic models, the icing process of wind turbine blades in the atmosphere can be numerically simulated. The model includes fluid flow simulation, droplet behavior, surface thermodynamics, and phase change. The airflow behavior was simulated by solving the nonlinear partial differential equations of mass, momentum, and energy conservation [19].

$$\frac{\partial \rho_\alpha}{\partial t} + \vec{\triangledown}\left(\rho_\alpha \vec{v_\alpha}\right) = 0 \tag{1}$$

$$\frac{\partial \rho_\alpha \vec{v_\alpha}}{\partial t} + \vec{\triangledown}\left(\rho_\alpha \vec{v_\alpha} \vec{v_\alpha}\right) = \vec{\triangledown} . \sigma^{ij} + \rho_\alpha \vec{g} \tag{2}$$

$$\frac{\partial \rho_\alpha E_\alpha}{\partial t} + \vec{\triangledown}\left(\rho_\alpha \vec{v_\alpha} H_\alpha\right) = \vec{\triangledown}\left(k_\alpha(\vec{\triangledown} T_\alpha) + v_i \tau^{ij}\right) + \rho_\alpha \vec{g} \vec{v_\alpha}, \tag{3}$$

where $\rho$ is the air density, $v$ is the velocity vector, subscript $\alpha$ is the air solution, $T$ is the static air temperature in Kelvin, $\sigma^{ij}$ is the stress tensor, and $E$ and $H$ are total initial energy and enthalpy, respectively.

Two-phase flow (air and water droplets) was numerically simulated by Euler's method, in which the supercooled water droplets were assumed to be spherical. Euler's two-phase fluid model consists of the $N - S$ equation and the droplet continuity and momentum equation. The droplet drag coefficient is based on the empirical correlation of flow around a spherical droplet described by Clift et al. [20]

$$\frac{\partial \alpha}{\partial t} + \vec{\triangledown}(\alpha \vec{V_d}) = 0 \tag{4}$$

$$\frac{\partial (\alpha \vec{V_d})}{\partial t} + \vec{\triangledown}\left(\rho\alpha \vec{V_d} \bigotimes \vec{V_d}\right) = \frac{C_D Re_d}{24k\alpha\left(\vec{V_a} - \vec{V_d}\right)} + \alpha(1 - \rho_\alpha/\rho_d)\frac{1}{(Fr^2)} \tag{5}$$

$$\rho_f \left[\frac{\partial h_f}{\partial t} + \vec{\triangledown}\left(\overline{V}_f h_f\right)\right] = V_\infty LWC\beta - \dot{m}_{evap} - \dot{m}_{ice}. \tag{6}$$

### 3.2.2. Icing Model

The energy conservation equation was used to characterize the complex icing process on the blade surface. When droplets hit the blade surface, some water droplets would evaporate and return, and some water droplets would freeze into ice and adhere to the blade surface [19]. The energy conservation equation can be expressed as:

$$\rho_f \left[\frac{\partial h_f c_f \tilde{T}_f}{\partial t} + \triangledown \cdot (v_f h_f c_f)\tilde{T}\right] = \left[c_f(\tilde{T}_\infty - \tilde{T}_f) + \frac{\|v_d\|^2}{2}\right]v^\infty \cdot LWC \cdot \beta -$$
$$L_{evap}\dot{m}_{evap} + \left(L_{fus} - c_s\tilde{T}\right)\dot{m}_{ice} + \sigma\varepsilon\left(T_\infty^4 - T_f^4\right) - c_h\left(\tilde{T}_f - \tilde{T}_{re}\right) + Q, \tag{7}$$

where $\rho_f$ is the fluid density; $c_f$ is the heat transfer coefficient of overflow water; $c_s$ is the heat transfer coefficient of solid; $c_h$ represents the heat transfer coefficient of the water film whose height is $h$; $\varepsilon$ represents the radiative heat transfer rate of solid; *sigma* is the Stephen Boltzmann constant ($\sigma = 5.67 \times 10^{-8}$W/m$^2\cdot$k$^4$); $L_{evap}$ is the energy coefficient of

the transformation process from liquid water to water vapor per unit mass; $L_{fusion}$ is the energy coefficient of the transformation process from liquid water to ice per unit mass; $\tilde{T}$ is the temperature of the control body; $\tilde{T}_{re}$ is the temperature of the fluid flowing back to the control body; $T_{\infty}$ indicates the reference temperature; $V_{\infty}$ indicates the reference speed; $Q_{icing}$ means the energy of frozen water.

### 3.3. Wind Turbine Integrated Calculation Method

The comprehensive analysis of wind turbines [21] was based on Kane's dynamic method (Kane and Levinson, 1985). Kane's equation of motion consists of two parts—generalized principal force and generalized inertia force: [22]

$$F_r + F_r^* = 0 (r = 1, 2, \dots, P),\tag{8}$$

where a set of rigid bodies of type $w$ is characterized by reference frame $N_i$ and centroid point $X_i$. The formula of generalized active $F_r$ and generalized inertial force $F_r^*$ is as follows:

$$F_r = \sum_{i=1}^{w} {}^E v_r^{X_i} \cdot F^{X_i} + {}^E \omega_r^{N_i} \cdot M^{N_i} (r = 1, 2, \dots, P)\tag{9}$$

$$F_r^* = \sum_{i=1}^{w} {}^E v_r^{X_i} \cdot \left(-m_i {}^E a_r^{X_i}\right) + {}^E \omega_r^{N_i} \cdot \left(-{}^E H^{N_i}\right).\tag{10}$$

It is assumed that, for each rigid body $N_i$, the center of mass of the particle $X_i$ is actively imposed. ${}^E a_i^{X_r}$ is the acceleration at the center of the particle $X_i$. ${}^E v_r^{X_i}$ and ${}^E \omega_r^{N_i}$ local linear velocity and angular velocity, respectively. ${}^E H^{N_i}$ is the time derivative of the angular momentum of rigid body $N_i$ about its center of mass $X_i$ in the inertial frame.

### 3.4. ILM-Power Loss Computing Theory

The calculation method of production loss caused by the icing of wind turbine blades is different, and the calculation result is different. While ILM [23] is a method proposed in IEA task 19 to assess production losses due to icing based on standard SCADA data provided by modern wind turbines. The schematic diagram of ILM is shown in Figure 7.

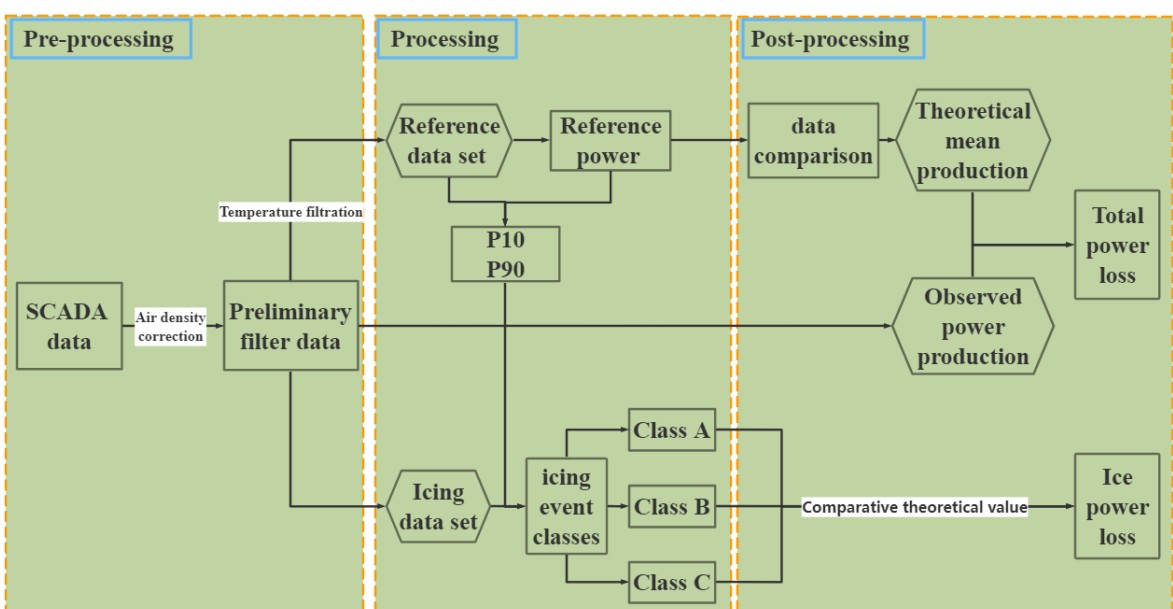

**Figure 7.** ILM schematic diagram.

In general, the method can be divided into three main steps [24]: Step 1: Calculate the reference power curve and the non-icing power curve; Step 2: Calculate the start and end timestamps of different icing event classes; Step 3: Calculate production losses due to icing.

The input data of this method mainly consist of wind speed, temperature, and power at different periods. The original data were initially screened and eliminated, and then the data were divided into two parts by temperature: clean and icing. The clean part was used to draw the reference power curve, the icing part was used to compare with the reference power to calculate the power loss caused by icing. A barometric correction is required in Step 1 to correct the air density to the hub height according to ISO 2533. Since turbine SCADA usually lacks barometric measurements, static pressure 1 was based on the site above sea level. Field air density and air pressure were used to calibrate engine room wind speed as follows:

$$w_{site} = w_{std} \times \left( \frac{\rho_{std}}{T_{site}} \right)^{\frac{1}{3}} = w_{std} \times \left( \frac{\frac{P_{std}}{T_{std}}}{\frac{P_{site}}{T_{site}}} \right)^{\frac{1}{3}} \tag{11}$$

$$w_{site} = w_{std} \times \left( \frac{T_{std}}{T_{site}} (1 - 2.25577 \times 10^{-5} \times h)^{5.25588} \right)^{\frac{1}{3}}, \tag{12}$$

where $w_{site}$ calibration engine room is the wind speed, $w_{std}$ is the measurement engine room wind speed, $T_{site}$ is the cabin temperature, $T_{std}$ is the standard temperature of 15 °C (288.15 K), air density is 1.225 kg/m$^3$, sea level $P_{std}$ = 101,325 Pa ambient pressure, and $h$ is the altitude.

The different icing events in Step 2 are divided into three types: Icing event class a: reduction of output power due to icing; Icing event class b: wind turbine stop due to icing; Icing event class c: anemometers are sometimes affected by ice, leading to overproduction.

The SCADA data of the measured wind turbine were from the wind turbine working record in a real complex environment. Because there are no measured data for the current 15 MW wind turbine, the method proposed in this paper uses WTIC to simulate the SCADA data and input the value of Icing event class A into the ILM method to filter out the icing power loss.

## 4. Result and Discussion

### 4.1. Grid Convergence Analysis

The NACA0012 airfoil was used for grid convergence analysis. Firstly, the feasibility of this method was verified by comparing the calculation and experimental results. The structural grid was drawn in ICEM, and the multi-shot icing method was adopted in Fensap-ice. The selection of the icing environment parameters is shown in Table 3 [25]. Figure 8 shows a comparison between the calculated results and wind tunnel test results, and it is found that the icing shape is consistent with the ice accumulation trend, which verifies the feasibility of the icing method.

**Table 3.** Enviromental parameters.

| Parameter | Value | Units |
|---|---|---|
| Temperature | −6.5 | |
| Wind speed | 63.6 | m·s$^{-1}$ |
| LWC | 0.77 | g·m$^{-3}$ |
| MVD | 15 | μm |
| Ice accumulation time | 546 | s |
| Angle of attack | 3.0 | deg |
| Characteristic length | 0.4572 | m |

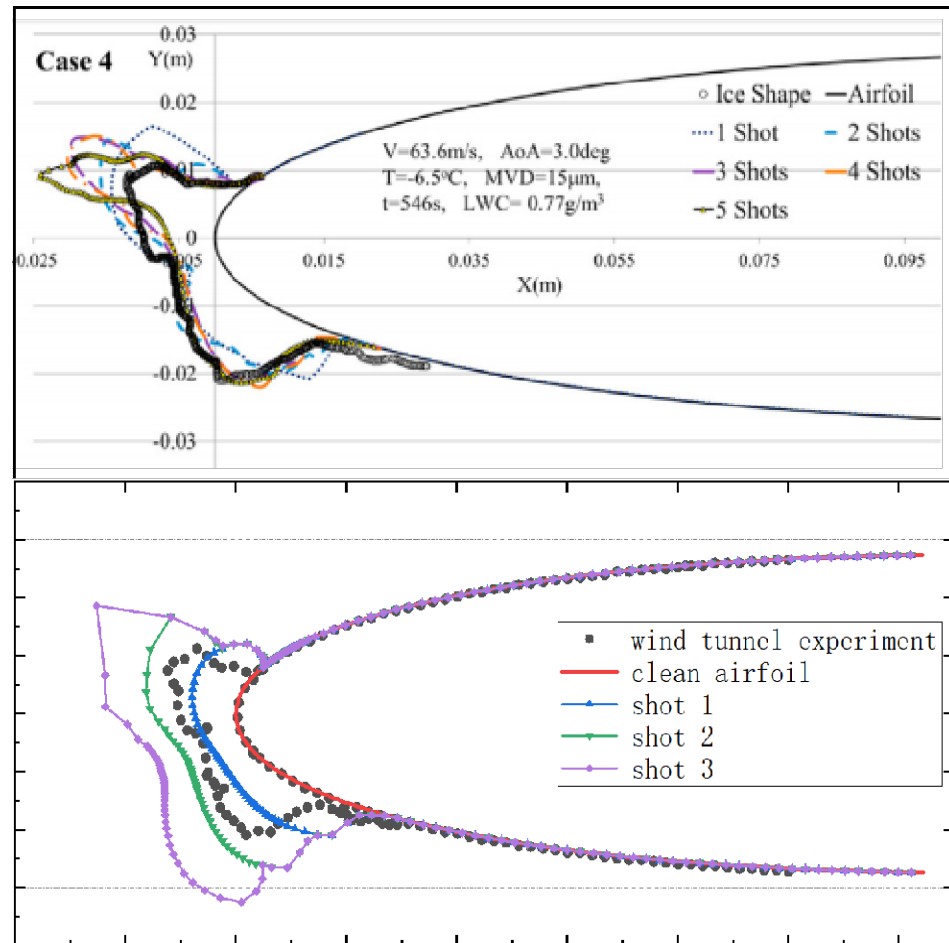

**Figure 8.** The NACA0012 airfoil wind tunnel test verification.

The grid independence test also adopts the multi-lens icing method, and a total of four groups of grids were set up for calculation and comparison. Figure 9 shows the comparison of the density of the front edges of the four groups of grids and the total number of grids. The average increase in the number of grids in each group is 25% to meet the verification requirements. Figure 10 shows the two-dimensional airfoil icing profiles corresponding to the four groups of grids. The results show that the icing effects of the four groups of grids are similar. To more clearly compare the icing differences between the four groups of grids, the three shots were divided and compared. Figure 11 shows the icing comparison of the three sub-shots of the icing of the four groups of grids with multiple shots. It is found that the icing shapes of the four groups of grids are almost the same in shot 1, and the outlines are slightly different in shot 2. The icing shape of shot 3 is different at the tip, but the overall icing trend is still the same. The reason for the difference is that each sub-shot of the multi-shot icing method will re-mesh and calculate the flow field based on the shape of the last icing. The irregularity of icing will lead to a slight deviation of the mesh partitioning results based on the density of the initial grid. Thus, the shape of the icing next time will be affected. Considering the calculation accuracy and efficiency comprehensively, grid 3 was selected in this paper for subsequent icing calculation.

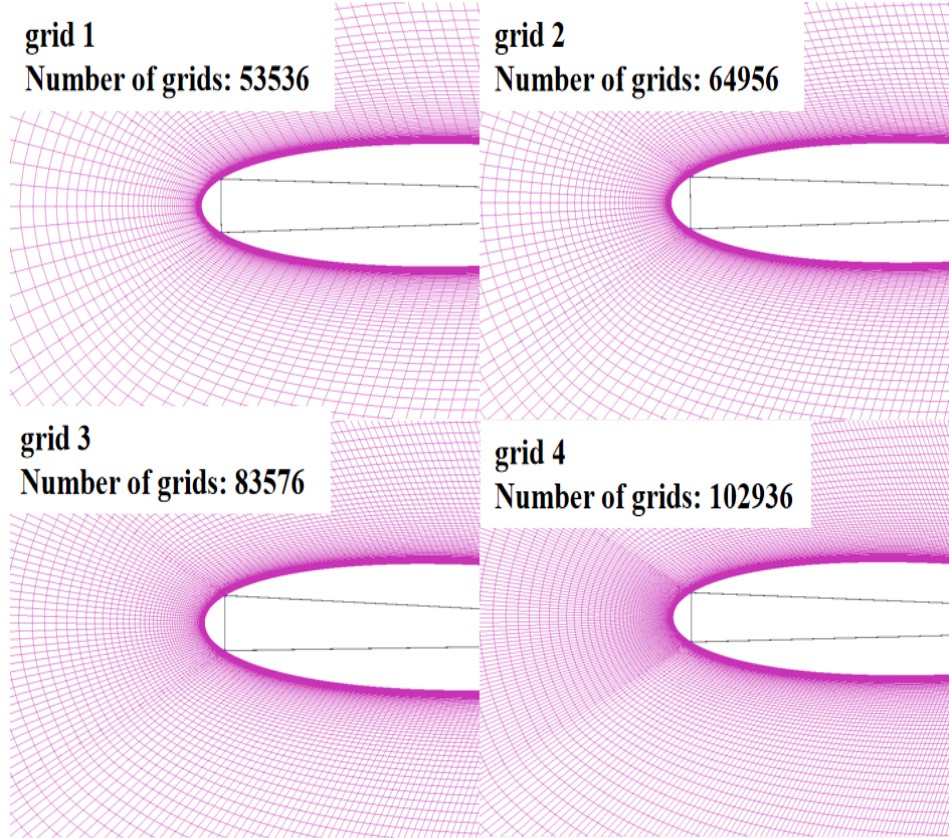

**Figure 9.** Comparison of four groups of grids.

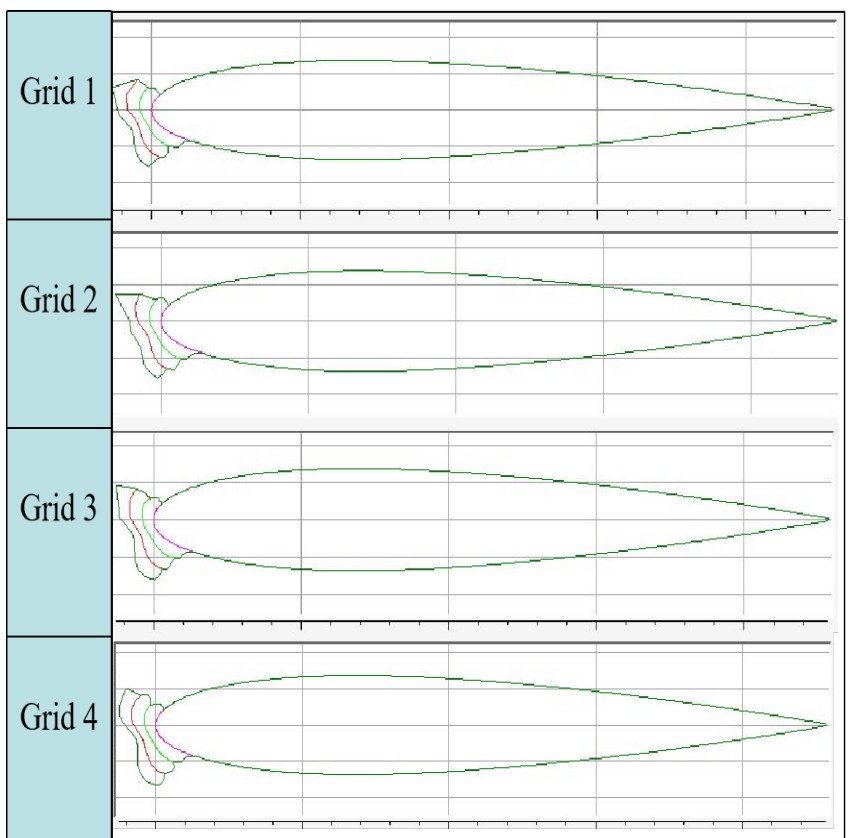

**Figure 10.** Grid 1–4 ice accumulation result.

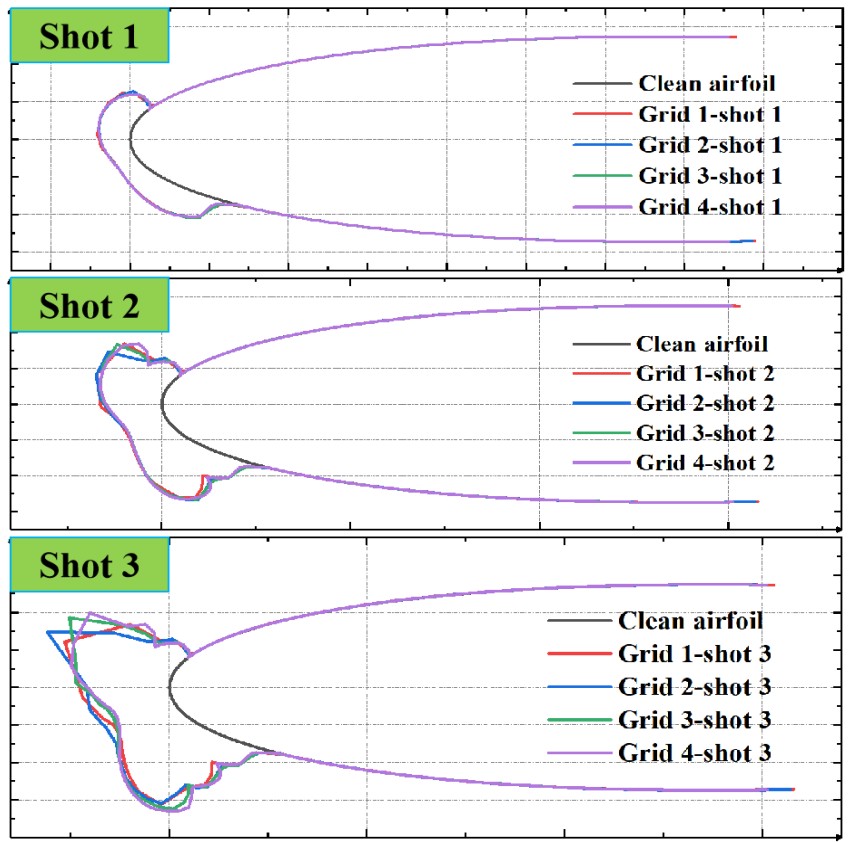

**Figure 11.** Shot 1–3 Comparison.

*4.2. Icing Discussion*

4.2.1. Ice Accumulation Process

In Catia software, the 13 selected blade sections were modeled, respectively, in the C-type fluid, and grid 3 was selected to divide the structural grid in ICEM software. The blade icing mainly occurs at the tip and has the greatest impact on output power. The airfoil distribution of 15 MW blades is 22.83% on the FFA-W3-211 airfoil near the tip. Therefore, we only selected the representative FFA-W3-211 airfoil at 95% of the section of the tip. The distributions of droplet movement, droplet velocity, ice accumulation position, and heat distribution before and after icing were compared.

Figure 12 shows the comparison of droplet movement distribution and droplet velocity distribution before and after icing. It can be seen from the water droplet motion distribution diagram that, after the airflow containing water droplets impinges on the leading edge of the airfoil, the airflow moves around the airfoil. The airflow direction has a certain angle-of-attack, so the water droplets' distribution on the pressure surface is closer to the airfoil surface than the suction surface, and the difference is more obvious on the trailing edge of the airfoil. By comparing the distribution of water droplets at the leading edge of the airfoil with that at the trailing edge of the airfoil, it is found that the contact between water droplets and the airfoil mainly occurs at the collision position of the leading edge of the airfoil, while the water droplets at the trailing edge have almost no contact with the surface of the airfoil, so the icing is mainly distributed at the leading edge of the airfoil. Comparing the distribution of water droplets before and after icing, it is found that the water droplet content decreases slightly. This is because the roughness of the water droplet collision area after icing will increase significantly compared with the clean airfoil, resulting in a significant increase in the capture efficiency of water droplets. It can be seen from the water drop velocity distribution diagram that the suction surface velocity of the airfoil is much higher than that of the pressure surface. The water drop velocity at the leading edge of the airfoil and the trailing edge of the pressure surface is lower, which makes it easier

for the water droplets to contact and form ice. Compared with the water drop velocity distribution before and after icing, the overall trend has almost no change and only slightly affects the movement of the highest velocity point before and after icing, which is caused by the shape of the airfoil changed by icing.

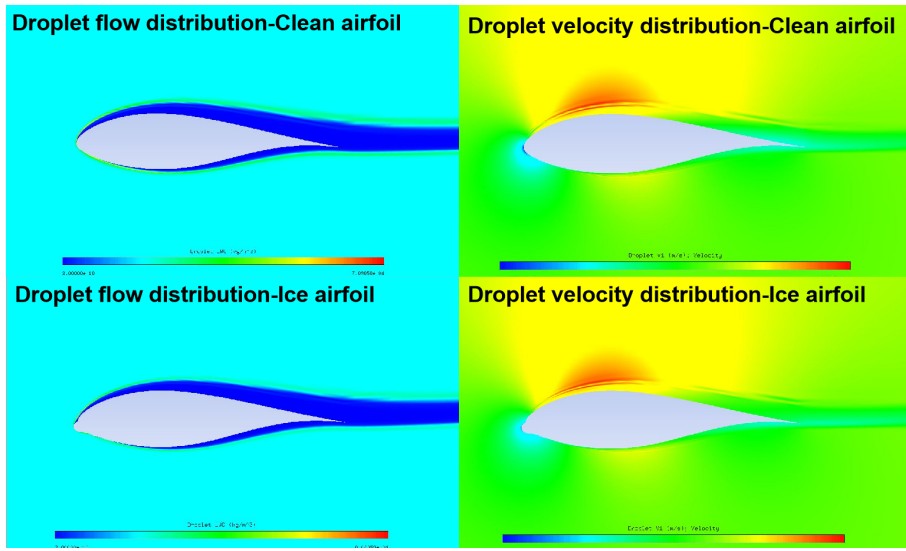

**Figure 12.** Droplet content and velocity distribution.

Figure 13 shows the comparison of ice accumulation position and temperature distribution before and after icing. It can be seen that the position of the ice mass caught is consistent with the distribution law of water droplet movement in Figure 12. The maximum water droplet collision occurs at the leading edge of the airfoil, and the ice accumulation effect will be more obvious. Compared with the ice mass caught distribution before and after icing, the contrast of the ice capture distribution of the airfoil after icing is greater, the distribution is concentrated at the ice tip, and the phenomenon of uneven distribution appears, which is caused by the unevenness of icing. The heat distribution in the figure is mainly concentrated on the pressure surface of the airfoil, which corresponds to the airflow velocity distribution in Figure 12. The heat is more concentrated in the area with low velocity. This is only a relative temperature distribution analysis and the maximum temperature is $-10.38\,^{\circ}\text{C}$, so the icing process will not be affected.

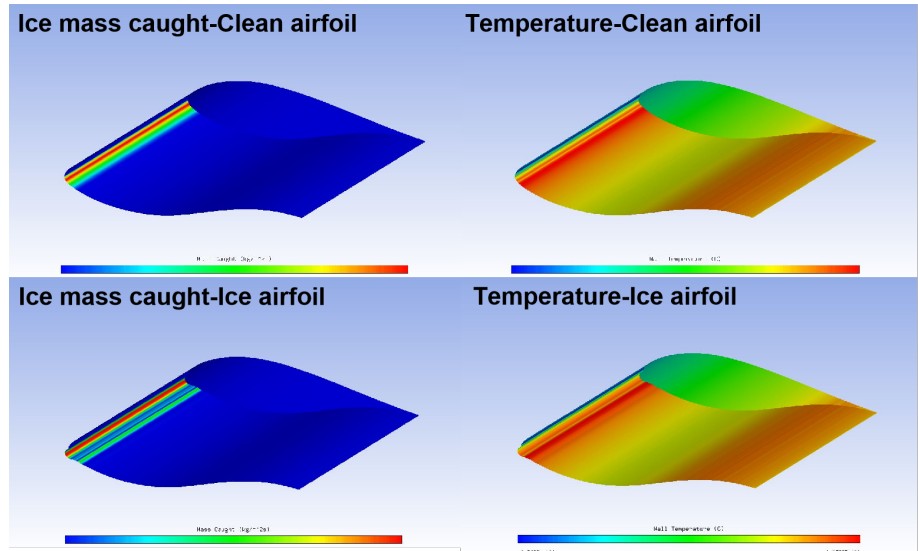

**Figure 13.** Icing and heat distribution.

### 4.2.2. Icing Effect

Figure 14 shows the icing situation of a single airfoil. It is consistent with the icing analysis characteristics mentioned above. It can be seen that the icing is mainly distributed on the leading edge of the airfoil, and the trailing edge will produce slight icing. Figure 15 shows the icing profile of the first eight sections of the selected tip. It is found that the amount of icing from the tip to the root of the blade gradually decreases, which is in line with the icing trend of the wind turbine blade in the actual situation. Blade icing is greatly affected by the relative velocity between the blade and air. The closer the blade is to the tip, the higher the linear velocity will be. As a result, the impact between the blade and water droplets will increase, resulting in more serious icing.

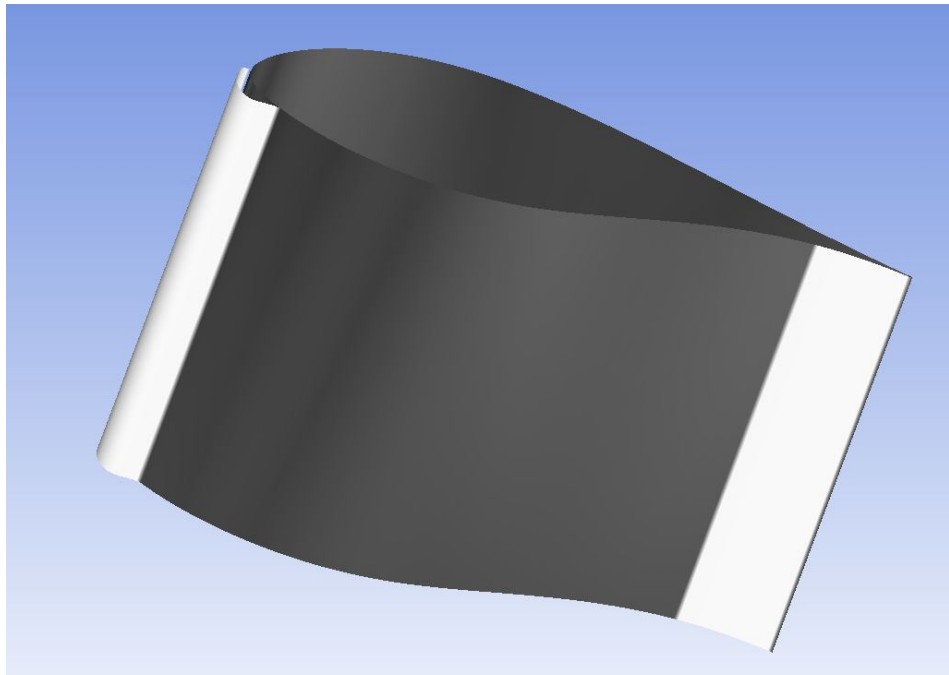

**Figure 14.** Ice accumulation location distribution.

### 4.2.3. Aerodynamic Characteristic Discussion

In the section on aerodynamic characteristics analysis, we also took the FFA-W3-211 airfoil at a 95% section of the blade tip as an example to analyze its aerodynamic characteristics. After the icing is completed in Fensap-ice, the grid after icing will be recalculated and generated as shown in Figure 16. The Spalart–Allmaras turbulence model will be adopted in Fensap-ice to calculate the generation drag coefficient. Therefore, the range of sweep air attack Angle set is $-10°$ to $20°$ and the angle-of-attack step is $1°$. However, in the calculation of wind turbine integration, the wind turbine can truly simulate the dynamic change of the wind turbine. The wind turbine control system will adjust the pitch of the blade appropriately according to the real-time wind speed and direction, to change the angle-of-attack of the blade airfoil, to ensure the safe operation of the wind turbine and achieve the best power generation efficiency. Therefore, the angle-of-attack range of the lift drag coefficient should be extended and modified to $-180°$ to $180°$ by using the Seling–Eggars method and the Viterna method [7].

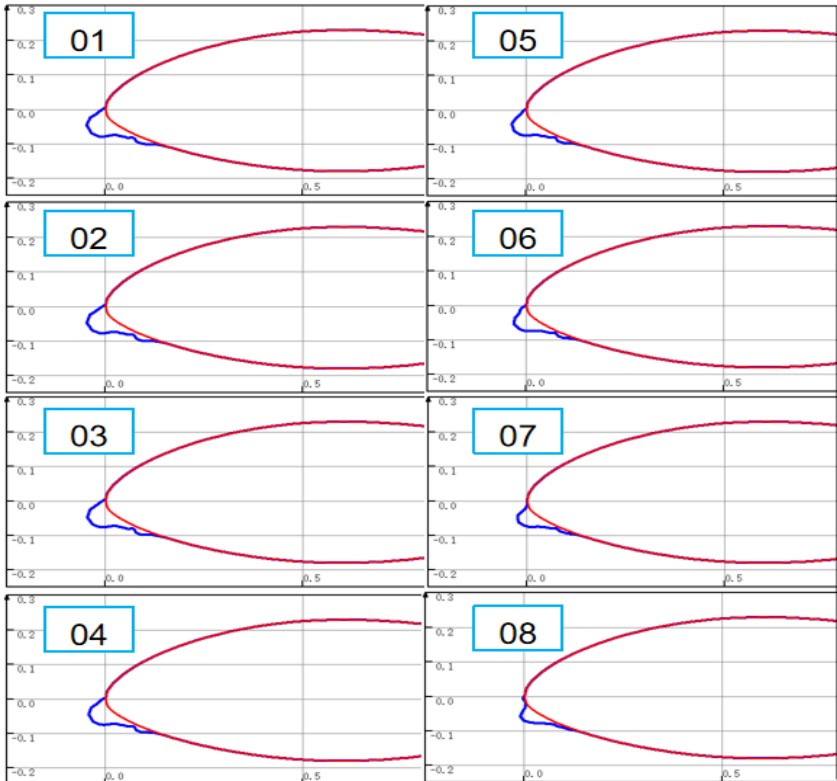

**Figure 15.** Icing effect of the first 8 sections of blade tip.

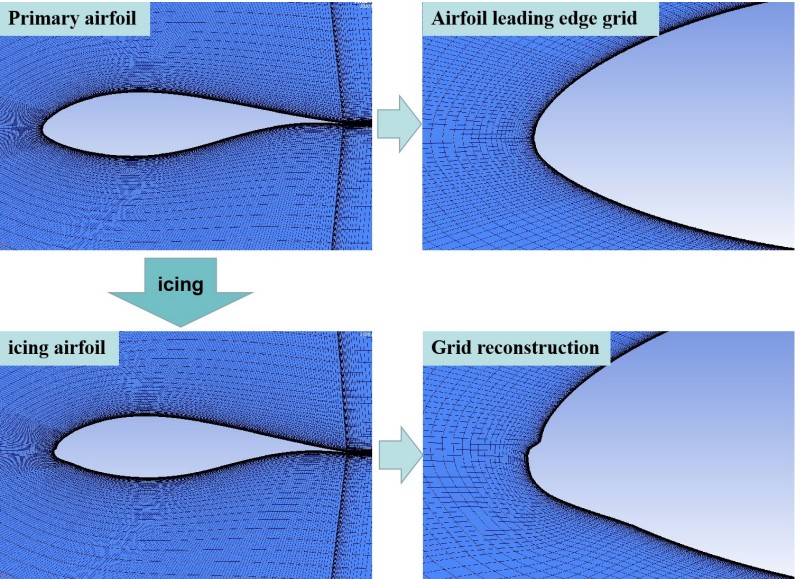

**Figure 16.** Fensap-ice grid reconstruction.

Figures 17 and 18 show the comparison between the clean blade lift coefficient and drag coefficient after icing, as well as that provided by NREL. When the angle-of-attack is 0°, the lift coefficient decreases by 24.32% and the drag coefficient increases by 165.47%. When the angle-of-attack is 90°, the lift vanishing drag reaches the maximum, the maximum lift coefficient of the iced airfoil is reduced by 41.19%, and the maximum drag coefficient is increased by 4.67%. It can be seen that the stall angle is advanced from 14.8° to 8° as a result of blade icing, which seriously affects the aerodynamic performance of the blade and will affect the power to a certain extent.

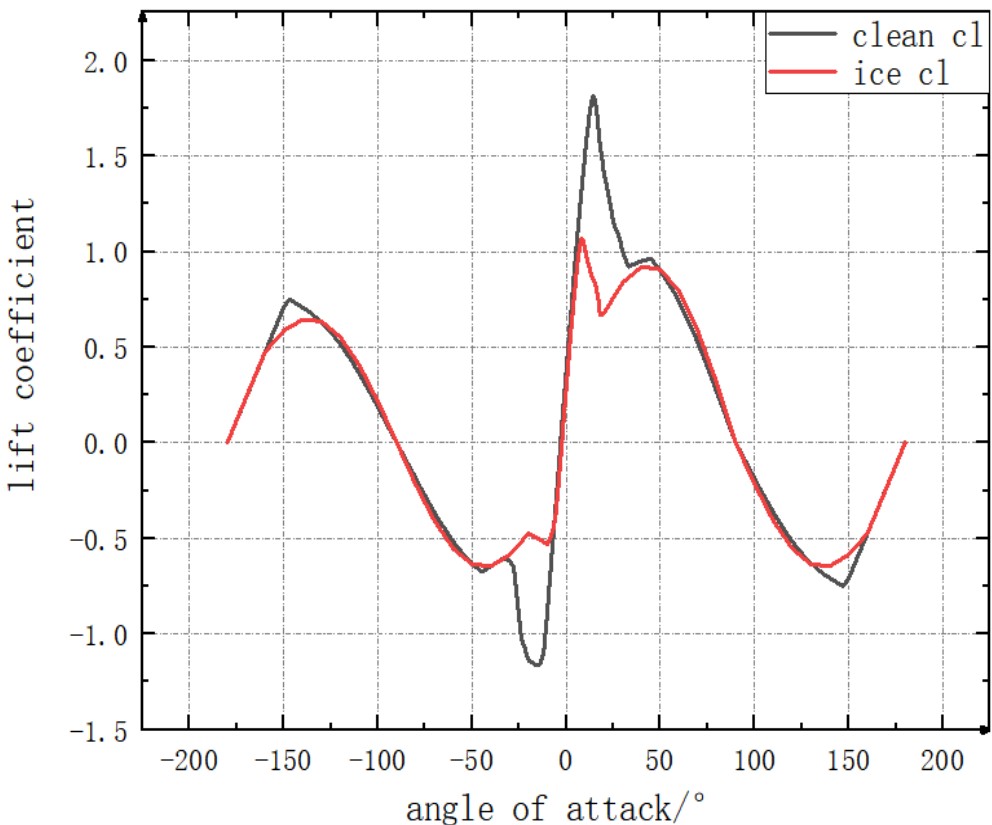

**Figure 17.** Lift coefficient of FFA-W3-211 airfoil.

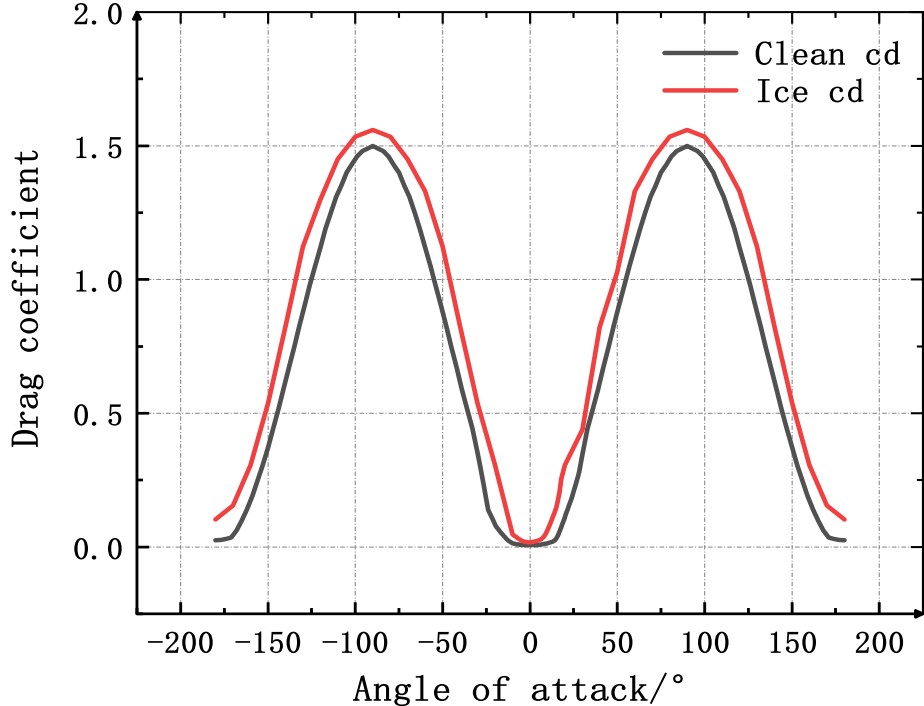

**Figure 18.** Drag coefficient of FFA-W3-211 airfoil.

### 4.3. Power Discussion

4.3.1. WTIC Result

The Wind Turbine Integrated Calculation needs to modify the original 15 MW clean blade model. The two-dimensional icing profile data and aerodynamic data of the blade calculated by CFD were transmitted to the integrated airfoil and aerodynamic modules of the wind turbine. We chose the most typical monopile wind turbine. The minimum distance between the wind turbine blade and the sea level is 30 m. The height of waves and fog is limited, and the influence of wind turbine blade icing can be ignored. Therefore, only the effect of wind load was considered, and the effect of fog caused by waves was not considered.

In the case of blade icing, to better analyze the dynamic response and power change of the wind turbine under different wind speeds, this paper took 1 m/s as the step length and calculated 24 working conditions from 3 m/s to 25 m/s, including the rated wind speed of 10.59 m/s. To compare the power and other performance parameters of the wind turbine during operation with clean blades, the clean blade also calculated the running characteristics of the wind turbine under different wind speed conditions.

Figure 19 shows the comparison of power curves before and after blade icing. The power curve is divided into three areas according to the wind speed from 3 m/s to 25 m/s. Region 1 is the wind speed of 3–6.98 m/s, which is the start-up stage of the wind turbine. The PI controller on the blade variable Angle adjusts the rotor speed to ensure that the minimum rotor speed is 5 rpm to start operation; Region 2 is the wind speed of 6.98–10.59 m/s, which is the optimization and adjustment stage of the wind turbine, to ensure that the rotor speed of the wind turbine is under the optimal tip ratio (TSR) when the wind speed is lower than the rated wind speed; Region 3 is the wind speed at 10.59–25 m/s, which is the phase when the wind turbine runs at rated power. When the actual wind speed is above the rated wind speed, the PI controller adjusts the rotor speed to its rated value of 7.55 rpm. It can be seen that icing causes the rated wind speed of the wind turbine to be delayed from 10.59 m/s to 13 m/s. When the wind speed is less than 13 m/s, the output power decreases obviously under the same wind speed condition. At the same time, it can be seen that, although icing occurs on the blade, the output power will also be generated when the wind speed reaches 3 m/s, indicating that icing will lead to output power reduction under the condition of low wind speed, rather than no output power generation such as a shutdown. When the wind speed is greater than 13 m/s, icing will no longer lead to the output power reduction. It can also achieve maximum power generation.

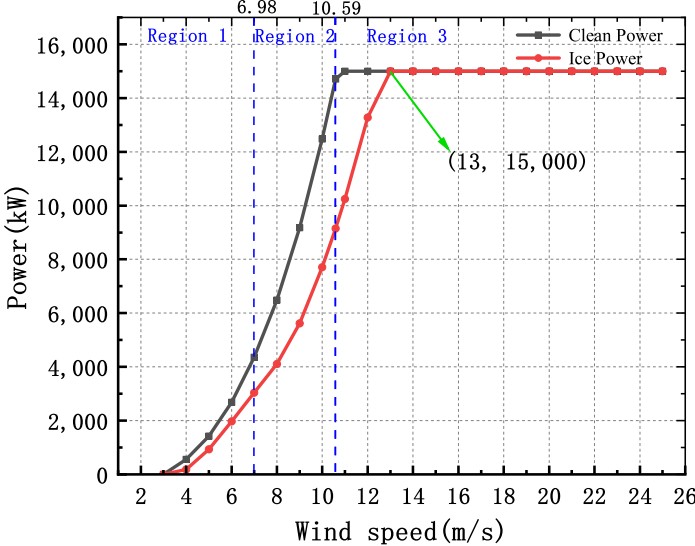

**Figure 19.** Comparison of power curves of clean and ice blades.

Figure 20 is based on Figure 19, which shows the power loss and power reduction ratio of the icing wind turbine running at 1 h under different wind speeds compared with that of the clean blade. It can be seen that, due to the small power base, the power loss ratio is the largest when the wind speed is 4 m/s, but the power loss is the largest near the rated wind speed, and the average hourly power generation under the rated wind speed is reduced by 5563.965 kW. The output power decreased by 38%; by integrating the power of rRegion 2 in Figure 17, it is found that the power loss of Region 2 is 37.48% due to icing.

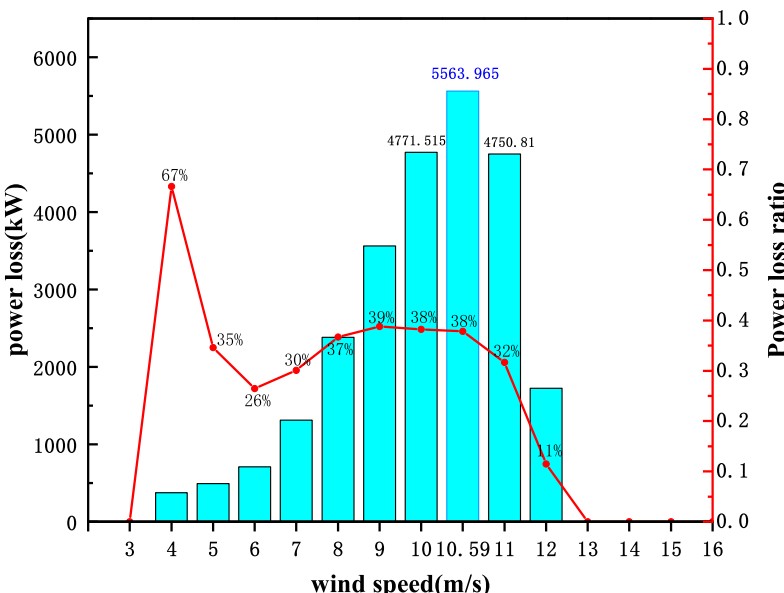

**Figure 20.** Comparison of power loss of different wind speeds.

4.3.2. ILM-Power Loss Discussion

In this paper, a method was proposed to simulate the SCADA data structure of the wind turbine by using the WTIC results of multiple groups of working conditions. The power loss caused by icing was analyzed by coupling the WTIC integrated analysis and the ILM method. In the integrated analysis of the wind turbine, multiple groups of data were calculated using 1 m/s step. Compared with the wind in the real environment, the working conditions of WTIC calculation were more stable. The data were collected after the steady operation of the wind turbine, and the simulated SCADA data set was relatively accurate and fixed, which is more conducive to calculating the power loss of the wind turbine caused by blade icing.

The requirement of the ILM method for the data set is that the reference power curve needs to be drawn first, and icing data can be compared. In this paper, two data structure simulations were carried out to compare the calculated power loss. Data set 1 was the clean blade power data, and data set 2 was the clean and ice mixed power data. The two sets of calculation results were compared to calculate the power loss under wind resource conditions with the wind speed increasing from 3 m/s to 20 m/s. Figure 21 and 22 show the two groups of power analysis charts, respectively calculated by ILM software and a statistical algorithm based on two sets of data. It can be seen that Figure 21 is the power curve of the clean blade and its trend is consistent with that of the clean blade in Figure 19. Figure 22 consists of the power curves of clean and icing parts, and it can be seen that the power curve of red icing decreases significantly between the wind speed range of 3 m/s and 13 m/s. The ILM method calculated that the power loss caused by icing accounts for 14.60% and the power loss calculated by wind turbine integration analysis is 12.65%. It can be seen that the calculation results of the two methods are similar.

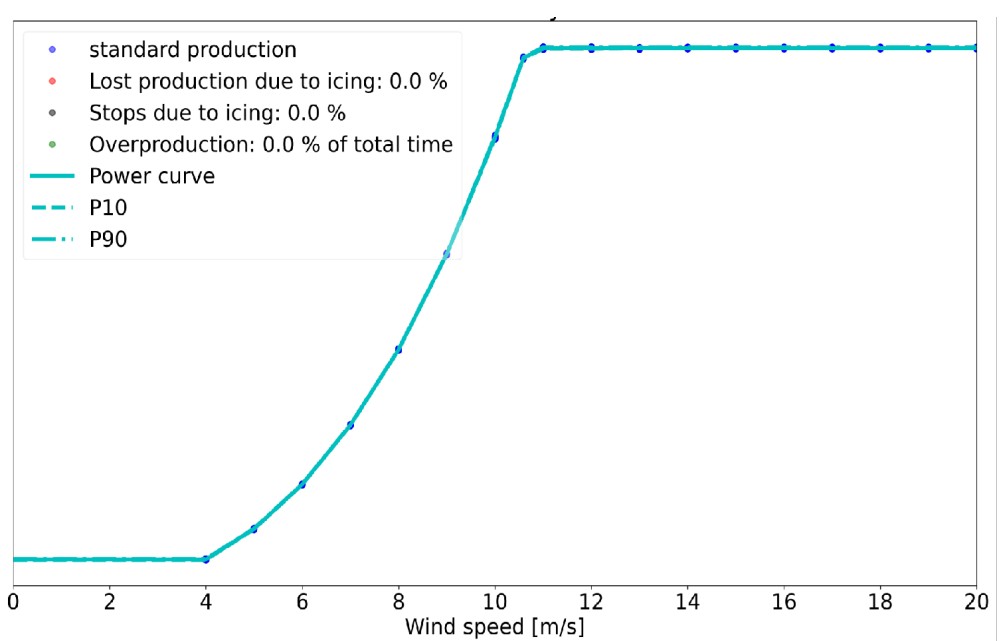

**Figure 21.** ILM clean power curve.

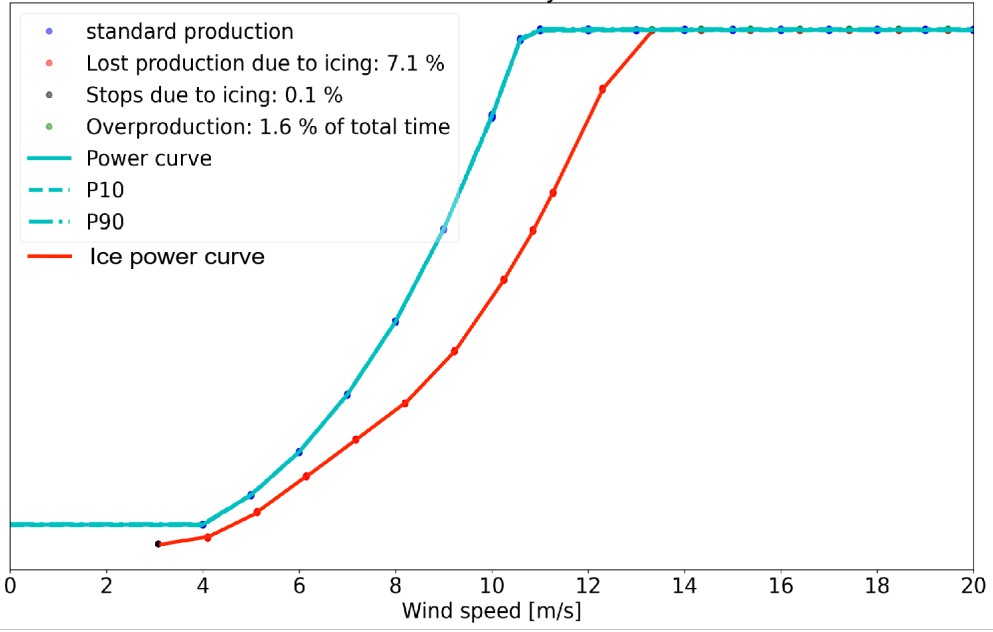

**Figure 22.** ILM clean and ice power curve.

In this paper, according to the real SCADA data case provided by task 19, the probability distribution data of wind resources were extracted, as shown in Figure 23, and SCADA data set 3 was simulated for one year to calculate the annual power loss caused by icing. Figure 24 is the comparison diagram of calculated output power. The calculation results are shown in Table 4. The annual theoretical power output value is 44,034,161.0 kWh, the actual power output value is 35,554,436.2 kWh, and the power loss caused by the icing of wind turbine blades accounts for 22.0%, and the annual loss value is 7,810,870.3 kWh. It can be seen that icing has a great influence on wind turbine power.

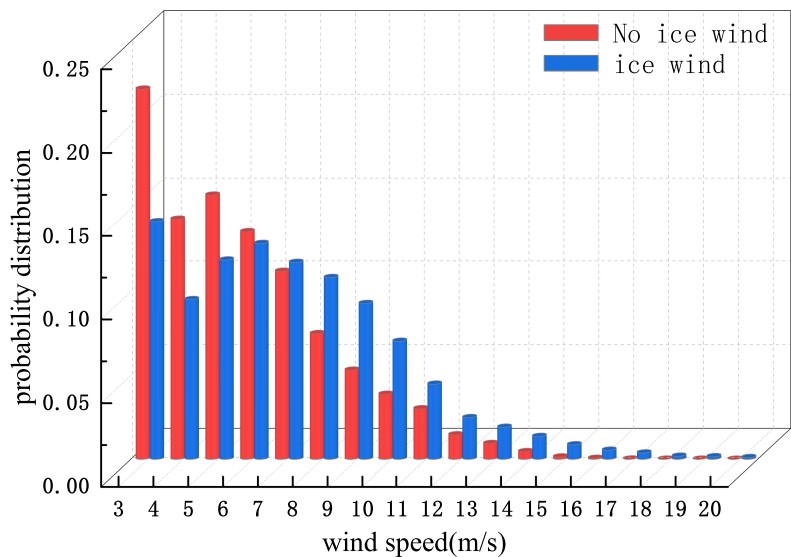

**Figure 23.** Probability distribution of wind resources in one year.

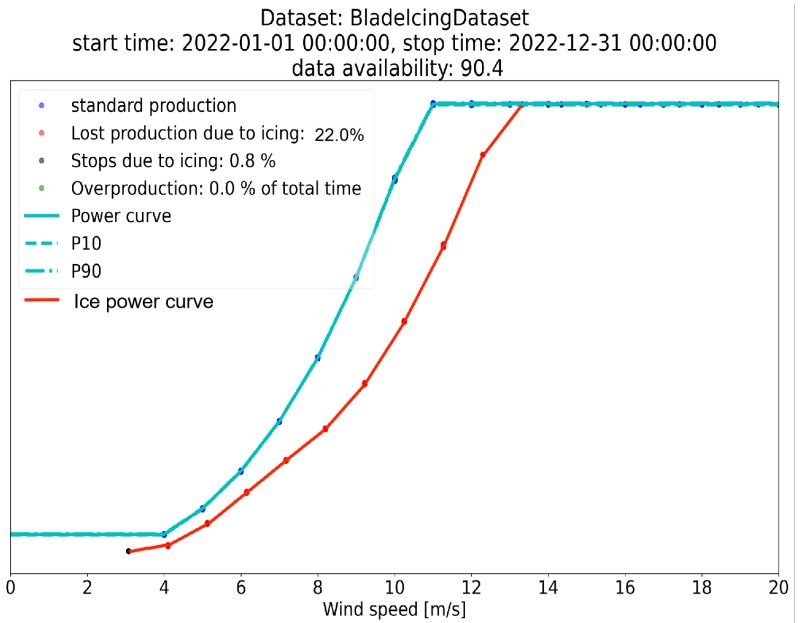

**Figure 24.** ILM Annual power loss calculation.

**Table 4.** ILM results.

| Project | Value | Units |
|---|---|---|
| Theoretical mean production | 44,034,161.0 | kWh |
| Observed power production | 35,554,436.2 | kWh |
| Icing during production | 3506.7 | h |
| Power loss ratio | 22.0% | - |
| Power loss due to icing | 7,810,870.3 | kWh |
| Downtime due to icing | 349,742.5 | kWh |

## 5. Conclusions

In this paper, the effect of blade icing on the power of the 15 MW wind turbine was studied. The CFD-WTIC-ILM multi-program coupling analysis method was proposed. The growth distribution law of two-dimensional blade icing on blades was analyzed, as well as the influence of icing on the output power of the whole machine during operation.

The power loss of the wind turbine caused by blade icing was analyzed using statistical methods and the conclusions were summarized as follows:

1.  It was found that the air flow will move around the airfoil after colliding with it. Icing is mainly distributed at the droplet impact site at the leading edge of the airfoil, and the trailing edge will produce slight icing. The icing is most obvious at the blade tip and the icing significantly decreases towards the blade root.
2.  It was found that the lift coefficient and drag coefficient change obviously before and after the airfoil icing at the tip. When the angle-of-attack is 0°, the lift coefficient decreases by 24.32%, the drag coefficient increases by 165.47%, the maximum lift coefficient decreases by 41.19%, and the maximum drag coefficient increases by 4.67%. Blade icing caused the stall Angle to push from 14.8° forward to 8°.
3.  The integrated analysis of wind turbine power found that icing causes the rated wind speed of the wind turbine to be delayed from 10.59 m/s to 13 m/s. When the wind speed is less than 13 m/s, the output power decreases under the same wind speed conditions. The average hourly power generation under rated wind speed decreases by 5563.965 kW, and the output power decreases by 38%. Secondly, icing causes the power loss of the Region 2 wind turbine in the optimization stage to reach 37.48%.
4.  Through ILM analysis, it was found that, under the condition of wind resources with equal spacing from 3 m/s to 20 m/s, the power loss caused by icing accounts for 14.60%, which is close to the power loss of 12.65% in the integrated analysis of wind turbines. By calculating the annual power loss caused by icing, it was found that the power loss caused by blade icing accounts for 22.0%, and the annual loss is 7,810,870.3 kWh. The degree of icing simulated in this study was small. The amount of icing will increase under more complex environmental loads and more serious power loss will be caused.

**Author Contributions:** Conceptualization, Z.C. and H.Y.; methodology, Z.C. and H.Y.; software, H.Y. and H.L.; validation, H.Y., H.L., H.Z. and L.X.; formal analysis, Z.C. and X.C.; investigation, H.Y.; resources, Z.C. and X.C.; data curation, H.Y.; writing—original draft preparation, H.Y.; writing—review and editing, Z.C. and X.C.; visualization, Z.C. and H.Y.; supervision, Z.C. and X.C.; project administration, Z.C. and X.C.; funding acquisition, Z.C. and X.C. All authors have read and agreed to the published version of the manuscript.

**Funding:** Basic Research Projects of Liaoning Provincial Department of Education: LJKMZ20220365.

**Conflicts of Interest:** The authors declare no conflict of interest.

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
