# Peer review of "Comprehensive Analysis of the Impact of the Icing of Wind Turbine Blades on Power Loss in Cold Regions"

_jmse, doi:10.3390/jmse11061125_

Round 1
Reviewer 1 Report
This is a very interesting study and a lot of work has been done. The literature study is comprehensive and up to date. Some comments are shown below:
1- page 1, line 15: ‘’ e rated power is pushed from 10.59 m/s to 13 m/s.’’ This will confuse the readers since the m/s is a velocity unit.
2- page 2, line 28: ‘’ measurement.Linyue Gao a, Tao Tao et Al’’. Remove the second author and stick to one author with adding et al.
3- ‘’BARBER’’, ‘’SIMO’’, ‘’KRAJ’’: why capital letters?
4- Check Ref [5], the author is not Barber, according to the list of references.
5- Please state clearly the novelty/contribution of your study in the last paragraph of the introduction, and compare that with what already have covered in the literature. What makes your study different?
6- page 2, line 88 to 90: please split the too long sentence.
7- abbreviations should be defined when mentioned for the first time such as WTC, MVD
8- Fig. 6: ‘’simulated’’ not ‘’simullated’’
9- There are some formatting problems such as ‘’5 rpmto’’. Please check the whole manuscript as I have noticed several such errors.
10- Fig. 17 and Fig. 19: Discussion is missing. You need to discuss the trends and observations and provide justifications rather than just interpreting the graphs into words.
Author Response
Dear teacher, thank you very much for your valuable advice. I have read it carefully and made corrections. Attached is the specific reply for your review.

Reviewer 2 Report
Dear Editor,
This paper titled “Comprehensive analysis of wind turbine blades icing impact on power loss in cold region” investigates the impact of blade icing on the power of a 15MW wind turbine, using a CFD-WTIC-ILM multi-program coupling analysis method. The growth distribution law of two-dimensional blade icing on blades, as well as the influence of icing on the output power of the whole machine during operation, are analyzed. The study finds that icing has a significant impact on the aerodynamic performance of the blade and the wind turbine's power output. The annual power loss due to blade icing is found to be 22.0%, which amounts to a loss of 7,810,870.3 kWh.
Finally, I would recommend the publication of this manuscript in Energies after the minor revisions are conducted based on the comments below.
Recommendations to improve the paper:
- Clarify the methodology: The paper would benefit from a more detailed explanation of the CFD-WTIC-ILM multi-program coupling analysis method. This should include a step-by-step description of the methodology, its advantages, and any potential limitations.
- The paper's literature review could be enhanced by providing a more thorough examination, which is crucial for emphasizing the study's innovative and unique aspects. To create a more exhaustive review, especially regarding the paper's emphasis on blade dynamics, it is recommended to include further references to structural research not presently cited in the introduction. Examples of these studies are: 10.3390/vibration4010001.
- Improve figures and tables: The quality of the figures and tables should be improved to enhance readability. This includes providing clear and concise captions, ensuring that the text within the figures is legible, and providing a clear, logical structure to tables.
- Discuss potential solutions: The paper could benefit from a discussion of potential solutions to mitigate the impact of blade icing on wind turbine performance. This may include techniques for ice detection, prevention, or removal.
- Address the impact of varying icing conditions: The paper should discuss the potential effects of different icing conditions on the results, including variations in the amount and type of ice, as well as the impact of different environmental conditions, such as temperature and humidity.
- Enhance the conclusion: The conclusion should be more comprehensive, summarizing the main findings, their implications, and suggestions for future research. Additionally, the conclusion should reiterate the study's contribution to the field of wind turbine performance and blade icing research.
- The English language quality of the text is generally good, with clear sentence structure and appropriate word choice. However, there are some minor grammatical errors and inconsistencies that could be improved upon. It is recommended to have the text proofread to ensure optimal clarity and readability.
Author Response

(The authors gave the same response as above.)

Reviewer 3 Report
The paper represents a really interesting investigation on a very difficult process like the icing of fan blades immersed in a cold humid air draft.
It would be important to improve the paper quality adding a deeper presentation of the icing process related to the flow kinematic impacting the blades. Mass, Entropy and Momentum conservation equations should be included and discussed.
Author Response
Dear teacher,
I have read your opinion carefully. In the previous article, the ice accumulation process was really less, which is an important part. Thank you very much for your valuable comments. I added "ice accumulation process" in 4.2.1 of this article. The distribution of air droplets and the capture of ice during the icing process are analyzed.
My sincere thanks.